# Suppression of pathological oscillations with transcranial focused ultrasound in Parkinson's disease

John Eraifej [1,2,3] ✉, Jake Toth[3], Jeremy Hanemaaijer[1,4], Shenghong He [2], Xinghao Cheng[3], Amir Puyan Divanbeighi Zand [1], Max E. Stewart[5], James J. FitzGerald[1,6], Christopher R. Butler [7,8], Timothy Denison[2,3], Alexander L. Green [1,6,9] & Robin O. Cleveland[3,9]

Transcranial ultrasound stimulation (TUS) is an emerging method for non-invasive neuromodulation of deep brain structures. However, to date, there is no evidence that TUS can directly modulate disease-related pathological oscillations in the same direction as known therapies. Inspired by clinical deep brain stimulation, in this randomised controlled cross-over study we probed the effects of pallidal TUS pulsed at 130 Hz on subthalamic beta-band activity, a biomarker in Parkinson's Disease (PD) in four male participants with PD. Beta-band power reduced in the ipsilateral subthalamic nucleus (STN) by 10.34% (95% CI:3.81% to 16.87%, p < 0.05, false discovery rate (FDR) adjusted). Beta power reduction was correlated between the ipsilateral ($R^2$ = 0.980, p < 0.05, FDR adjusted), but not contralateral, STN and primary motor cortex. Brady-kinesia, as measured by change in reaction time, was also reduced by 17.70% (95% CI:8.95% to 26.41%, p < 0.05, FDR adjusted). In this proof of concept study, we demonstrate that TUS can suppress pathological oscillations, potentially opening the door for therapeutic TUS (NCT06932185).

Parkinson's disease (PD) is a common neurodegenerative disorder characterised by the loss of dopaminergic neurons leading to brady-kinesia. Bradykinesia is the pre-requisite for clinical diagnosis when present alongside other symptoms such as resting tremor, rigidity or postural instability[1]. Dopamine depletion, and the subsequent disruption of basal ganglia network dynamics, is thought to be a key mechanism by which symptoms arise[2]. Therefore, the mainstay of pharmacological treatment is dopamine replacement therapy[3]. When patients suffer from the side-effects of increasing dosages or motor fluctuations, deep brain stimulation (DBS) is considered[4].

DBS, an established therapy for PD, involves implanting electrodes into either the subthalamic nucleus (STN) or the globus pallidus internus (GPi)[5]. DBS' adoption has provided an opportunity to record local field potentials (LFPs) from these deep brain nuclei, transforming our understanding of PD circuit pathology and biomarkers of disease[6]. Beta-band (13–30 Hz) power is elevated within both the STN and the GPi in PD, reduced by both dopaminergic medication and DBS[7,8], and correlated with symptom severity[9]. Notably, this modulation in beta-band activity can be detected within distributed neurophysiological networks[10]. Such is the reliability of this pathological oscillation as a

[1]Nuffield Department of Surgical Sciences, John Radcliffe Hospital, University of Oxford, Oxford, UK. [2]Brain Network Dynamics Unit, Nuffield Department of Clinical Neurosciences, John Radcliffe Hospital, University of Oxford, Oxford, UK. [3]Institute of Biomedical Engineering, Department of Engineering Science, University of Oxford, Oxford, UK. [4]Department of Neurosurgery, Radboud University Medical Center, Radboud University, Nijmegen, The Netherlands. [5]Nuffield Department of Orthopaedics, Rheumatology, and Musculoskeletal Sciences, University of Oxford, Botnar Research Centre, Oxford, UK. [6]Nuffield Department of Clinical Neurosciences, John Radcliffe Hospital, University of Oxford, Oxford, UK. [7]The George Institute for Global Health, Imperial College London, London, UK. [8]Department of Brain Sciences, Imperial College London, London, UK. [9]These authors contributed equally: Alexander L. Green, Robin O. Cleveland. ✉e-mail: john.eraifej@ndcn.ox.ac.uk

**Table 1 | Participant characteristics**

| Patient | Age | Sex | Dominant hand | Symptom dominance | Months since surgery | LEDD | Left Electrode | | | Right Electrode | | |
|---|---|---|---|---|---|---|---|---|---|---|---|---|
| | | | | | | | Amplitude (mA) | Pulse Width (µs) | Frequency (Hz) | Amplitude (mA) | Pulse Width (µs) | Frequency (Hz) |
| 1 | 54 | M | right | right | 3 | 612.5 | 1.6 | 50 | 125 | 1.4 | 50 | 125 |
| 2 | 63 | M | right | right | 5 | 620.0 | 1.7 | 50 | 130 | 2.2 | 50 | 130 |
| 3 | 55 | M | right | right | 3 | 667.5 | 0.8 | 50 | 125 | 1.3 | 50 | 125 |
| 4 | 54 | M | right | right | 5 | 546.2 | 1.5 | 50 | 130 | 1.3 | 50 | 130 |

LEDD levodopa equivalent daily dose[3].

biomarker of PD symptom severity, it is the basis for closed-loop DBS, a clinically approved therapy where DBS is only activated when elevated beta-band activity is detected[11,12].

Transcranial ultrasound stimulation (TUS) is emerging as an exciting new method of non-invasive neuromodulation due to its ability to target deep brain structures with high spatial resolution[13]. TUS has therefore been proposed as a potential therapy[14]. Whilst promising therapeutic effects have been reported after TUS in several conditions including depression[15], chronic pain[16] and essential tremor[17], these studies have either been open label[15,17] or relied on subjective self-reporting[15,16]. To date there has been no evidence that TUS can directly ameliorate pathological oscillations associated with disease. This lack of evidence may be partly explained by technical challenges and a lack of robust biomarkers in the conditions tested, unlike in PD. Furthermore, the optimal parameter set for therapeutic TUS is difficult to identify given the large parameter space available and the differential site-specific effects[18]. In contrast, DBS for PD has used electrical stimulation pulsed at around 130 Hz to improve bradykinesia for over 30 years[19].

In this randomised controlled proof-of-concept study, we leverage PD as a model system and show that DBS-inspired 130 Hz pulsed GPi-TUS reduces STN beta-band power. We also show that this reduction is coupled to a distributed network that includes the motor cortex and that GPi-TUS improves bradykinesia, as measured by reduction in median reaction time.

Four consecutive PD patients with STN-DBS using the Medtronic Percept® device, which allows for telemetric LFP recordings, were recruited (Table 1). Based on pre-operative diffusion tensor imaging (DTI), a peak STN to GPi connectivity region was identified within the left GPi and was set as the TUS target (Fig. 1A). Participants received TUS with the same pulse duration (90µs) and pulse repetition frequency (PRF; 130 Hz) as is used in DBS (Fig. 1B). This was applied to either the left GPi or left ventricle on consecutive days in a randomised order, to which the participant and clinical assessor were blinded. To account for the known diurnal variation in beta power[20], experimental days consisted of a sham block followed by a TUS block (Fig. 1C). To reduce the risk of the auditory artefact associated with pulsed TUS an acoustic masking signal was played through headphones during all experiments[21]. All TUS block electrophysiological and behavioural data were normalised to the sham block and strict timing was maintained between days. All experiments followed an 'on-line' protocol, where LFP, EEG and behavioural data were collected during TUS at rest and during two behavioural tasks. All sham and TUS experiments were completed between dopamine replacement therapy doses. Abbreviated Unified Parkinson's Disease Rating Scale part III (UPDRS-III) assessments were completed at the end of each block (Fig. 1C).

## Results
### Personalised acoustic lenses focus TUS to deep brain nuclei
Target-specific personalised acoustic lenses were designed using treatment planning software to focus ultrasound either to the GPi or to the frontal horn of the left lateral ventricle as an active control site[22], whilst avoiding the STN electrode. This allowed us to probe network dynamics whilst minimising the risk of potential artefacts associated with sonicating the electrode[23]. In each participant, the GPi target was personalised according to the peak structural connectivity from the STN to the posteroventral GPi using established DTI pipelines. Personalised acoustic lenses were able to focus TUS to the entry point within the GPi of these STN-GPi white matter tracts (Fig. 2). Each lens was validated by comparing measurements in water with the predictions from the treatment planning software (Fig. S2).

### 130 Hz GPi-TUS suppresses beta oscillations in the STN
GPi-TUS led to a relative reduction in ipsilateral STN mean beta power in all four participants at rest (Fig. 3A). Strikingly, where distinct low (13–20 Hz) and high (21–30 Hz) beta peaks were present in a participant's power spectrum, only low-beta power was reduced (Fig. 3B). At the group level, GPi-TUS led to a mean relative reduction in ipsilateral STN beta power of 10.34% (95% CI: 3.81–16.87%, $p < 0.05$, false discovery rate (FDR) adjusted) when compared to Ventricle-TUS. Although there was a reduction in contralateral STN beta power, this did not reach statistical significance (Fig. 3B).

### GPi-TUS has specific network-level effects
GPi-TUS also led to networked modulation of cortical beta activity. As expected from the literature[10], there was no significant reduction in mean beta power in the ipsilateral (left) and contralateral (right) primary motor cortices at the group level (M1, Fig. 3C), however, mean beta power reduction in the ipsilateral M1 region correlated strongly with ipsilateral STN beta reduction ($R^2 = 0.980$, $p < 0.05$, FDR adjusted). In contrast, in the contralateral (right) hemisphere, STN beta reduction was not correlated with M1 beta reduction, indicating that GPi-TUS led to a lateralised network-level modulation in beta activity (Fig. 3E). There was no significant change in M1-STN beta coherence at the group level (Fig S3).

### 130 Hz GPi-TUS improves reaction time
GPi-TUS was also associated with a statistically significant reduction in median reaction time of 17.70% (95% CI: 8.95–26.41%, $p < 0.05$, FDR adjusted) compared to ventricle-TUS on the random dot-motion task. This improvement in bradykinesia was seen in all participants (Fig. 3F). Total abbreviated UPDRS-III scores were not significantly reduced at the group level (Fig. 3G).

### Beta suppression is not the result of mechanical TUS artefacts
To ensure that beta reduction was not the result of an artefact induced by off target TUS to the DBS electrode tip, we characterised the mechanical artefact associated with direct sonication of a DBS electrode. To do this, we inserted a DBS electrode into an agar phantom and sonicated the electrode tip directly using our experimental parameters in a water tank (Fig. S5). We observed no effect of the 130 Hz PRF at 600 kPa on electrode recordings (Fig. 4). Since the PRF used in

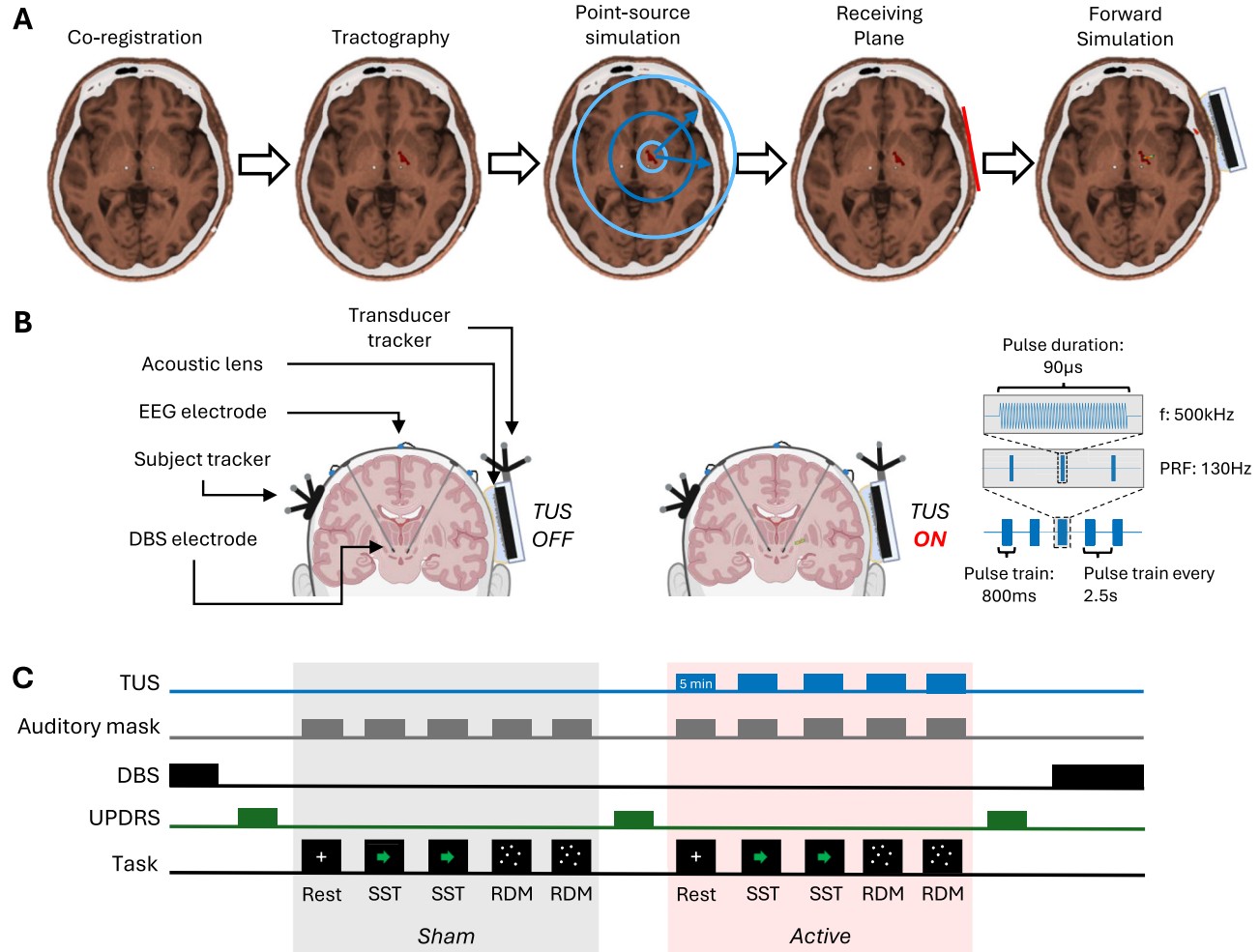

**Fig. 1 | Study design. A Pipeline for designing participant and target-specific acoustic lenses**: Pre-operative T1w MRI scans were co-registered to post-operative CT scans. Probabilistic tractography was used to find the peak STN-GPi connectivity location within the GPi and used as the source in a point-source simulation. Phase of ultrasound waves at the receiving plane was then corrected for with a PDMS lens and a forward simulation confirmed targeting of the GPi using the PDMS lens. **B Experimental set-up:** including Medtronic Percept DBS leads implanted in the subthalamic nuclei, neuro-navigation trackers on the participant and transducer, EEG electrodes and the custom PDMS lens. Created in BioRender. Toth, J. (2026) https://BioRender.com/cqg28ia. TUS was delivered at 500 kHz with 90 μs pulse duration at 130 Hz pulse repetition frequency (PRF) over a 800 ms pulse train duration. A 1.7 second delay allowed for cooling before the next pulse train was applied. **C Experimental protocol:** On each study day, participants underwent sequential sham and active TUS blocks with DBS turned off and an auditory mask. Unified Parkinson's Disease Rating Scale part III (UPDRS-III) assessments were performed at baseline, after sham and after active TUS. Active TUS targeted either the globus pallidus internus (GPi) or ventricle, in a randomised order. TUS was applied in 5 minute blocks at rest, during a stop signal task (SST) and during a random dot-motion (RDM) task. STN subthalamic nucleus, GPi globus pallidus internus, PDMS polydimethylsiloxane.

our study is above the hardware low-pass filter that is set at 100 Hz in the Percept® device[24], we sought to characterise the mechanical artefact of a PRF below this cut-off in order to confirm that a mechanical artefact could be detected, if present. We found that TUS delivered at 5 Hz PRF resulted in a visible signal at 200 kPa, with an amplitude above what would be expected for physiological signals (Fig. 4). At 5 Hz PRF, an artefact was visible in both the time (Fig. 4A) and frequency (Fig. 4B) domain. To confirm that 130 Hz PRF would not interact with the physiological frequency band of interest, we quantified the effect of direct electrode sonication on beta band power, again with 5 Hz PRF as a positive control. In the left electrode, beta-band power increased with 5 Hz and scaled with peak pressure but this was not true at 130 Hz (Fig. 4C). Whilst these results confirm that our findings cannot be explained by mechanical artefacts, we also ensured that the ultrasound focus was >5 mm away from the electrode tip to further reduce the risk of polluting the LFP signals and also to minimise artefactual heating in the vicinity of the electrode. Our in-silico simulations indicated a mean maximum temperature rise of $0.47 \pm 0.37$° C, within International Transcranial Ultrasonic Stimulation Safety and Standards (ITRUSST) consortium guidelines ($< 2$° C). Mean targeting error was $0.50 \pm 0.33$ mm for the GPi and $0.51 \pm 0.37$ mm for the ventricle (Table S2).

## Discussion

The present study demonstrates that TUS can suppress pathological oscillations in PD, causally and in real-time. We demonstrate that these effects are analogous to those seen with electrical DBS to the same target and that TUS, pulsed at the same frequency as DBS, leads to a reduction in STN beta power and a clinically relevant improvement in motor performance. To our knowledge, this is the first time that TUS has been shown to modulate a clinically established biomarker of disease in the same direction as a known therapy. In addition, in two participants, we also see that only low-beta power was reduced where both a low-beta and high-beta peak were present. Based on previous

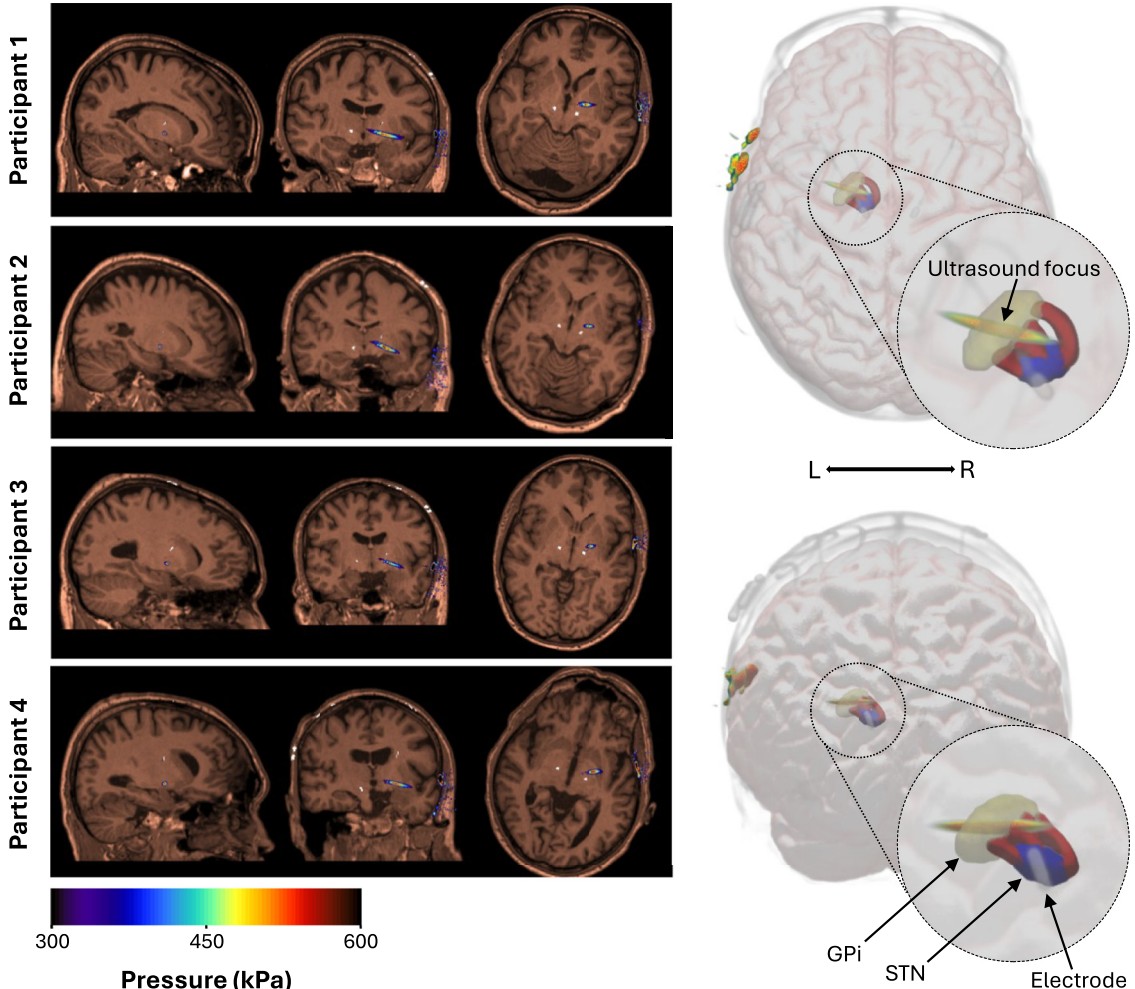

**Fig. 2 | Pressure maps in brain.** Simulated acoustic pressure fields in brain across all four participants. Simulations were performed using the k-Wave toolbox with a 500 kHz fundamental frequency. The focal region is visualised for each participant −6 dB (left) relative to the peak pressure in brain tissue. The maximum pressure on the colour scale (600 kPa) reflects the maximum pressure in brain tissue; higher pressure regions in the skull may saturate this scale and appear black. Note that the participant's DBS electrode can be seen in white. An example of a tractography-based ultrasound focus using a personalised acoustic lens is shown in three dimensions (right). The ultrasound focus can be seen overlying the entry point of STN-GPi tracts *(red)* whilst avoiding the participants DBS electrode. GPi globus pallidus internus (yellow), STN subthalamic nucleus (blue).

work, this would be in keeping with our targeting of a key output node of the basal ganglia[25] and highlights the potential for TUS to probe network dynamics non-invasively.

Whilst the exact effects of TUS on network dynamics remain largely unknown, the results of previous DBS studies may help explain our results. In the classical circuit model of PD, the basal ganglia comprises a direct, indirect and hyperdirect pathway[26]. Activation of the direct pathway promotes movement through net inhibition of the GPi (and substantia nigra pars reticulata), whilst activation of both the indirect and hyperdirect pathway inhibit movement and elevate circuit-level beta synchrony[26]. Low-beta synchrony is thought to arise from indirect pathway overactivity whilst high-beta synchrony arises from hyperdirect overactivity[25], although this may also contribute to low-beta synchrony[27]. Our observation that low-beta power was selectively suppressed by TUS (where both a low- and high-beta peaks were recordable) is in keeping with the literature since the GPi is the key output node of the indirect pathway and is bypassed by the hyperdirect pathway, a monosynaptic cortico-subthalamic projection. Similarly, we found a correlated reduction in ipsilateral M1-STN beta power without significant modulation in M1-STN coherence. Our results would therefore suggest that TUS, pulsed at 130 Hz, has an inhibitory effect on the GPi leading to inhibition of this overactive resonant network at the macroscale.

At the microscale, the direct cellular effects of TUS are less clear. Numerous mechanisms of action have been proposed including cell membrane capacitance modulation[28], mechanosensitive ion channel actuation[29–31], and synaptic neurotransmitter release[32,33] (for review see Blackmore et al.[13]). Any one of these potential mechanisms could yield a net inhibitory effect at the macroscale. For example, it is possible that changes in cell membrane capacitance within the GPi alter synaptic integration, action potential propagation speed, and firing frequency[34], thereby reducing the likelihood of low-beta synchrony propagation at the network level. It is also possible that actuation of mechanosensitive ion channels lead to an informational lesion within the GPi, as is observed with electrical stimulation elsewhere[35,36]. Since the majority afferent fibres to the GPi from the striatum are inhibitory[37], this could occur through synaptic neurotransmitter release which would have a net inhibitory effect within the GPi. Indeed, this is a proposed mechanism for the treatment effects of GPi-DBS[38,39]. Whilst these theoretical explanations of how the mechanical effects of ultrasound may translate into altered neurophysiology are speculative, they demonstrate that any one (or more) of the putative TUS

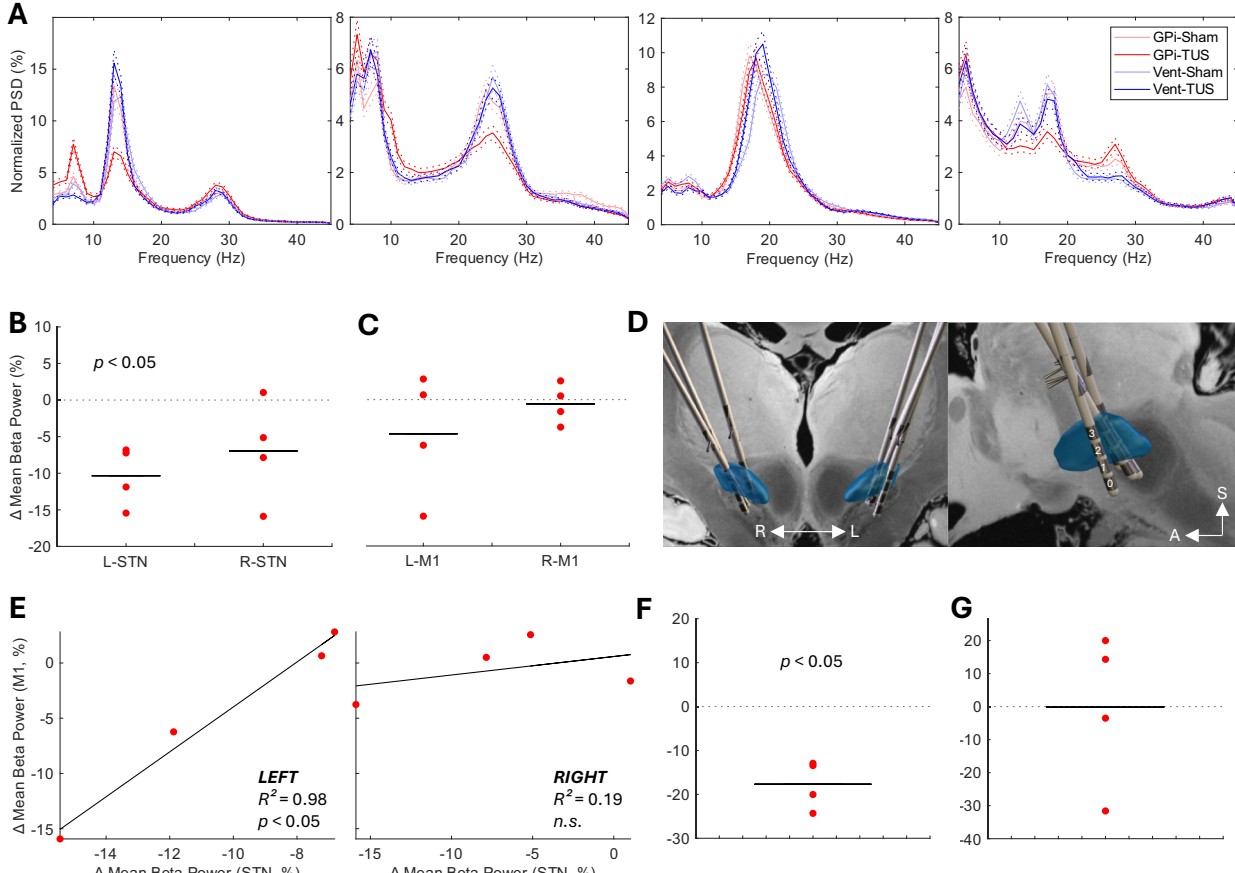

**Fig. 3 | Suppression of pathological oscillations and the associated behavioural effects. A Normalised power spectral density (PSD) in the subthalamic nucleus (STN) for each participant and condition. B Normalised mean beta power difference in the STN:** Group mean beta power difference (GPi-TUS − Ventricle-TUS) in the left and right STN (two-sided t-test, FDR adjusted *p* value = 0.04). **C Normalised mean beta power difference in the primary motor cortex (M1):** Group mean beta power difference between GPi-TUS and Ventricle-TUS in the left and right M1 region. **D Electrode reconstruction:** Electrode reconstruction of all four participants in MNI space[58]. Electrode numbering convention is shown for reference. *Blue = STN* **E) Network-level reduction in beta power during to GPi-TUS:** Correlation between M1 beta power reduction and STN beta power reduction in the ipsilateral (left) hemisphere (left panel, linear regression, FDR adjusted *p* value =

0.04) and the contralateral (right) hemisphere (right panel, linear regression, FDR adjusted *p* value = 0.76). **F Change in reaction time:** Difference in reaction time relative reduction compared to sham (GPi-TUS − Ventricle-TUS) during the random dot-motion task (two-sided *t* test, FDR adjusted *p* value = 0.04). **G Normalised abbreviated Unified Parkinson's Disease Rating Scale part III (UPDRS-III) difference:** Difference presented (GPi-TUS − Ventricle-TUS). Horizontal lines represent the mean and each red dot represents data from a single participant. Each active TUS block was normalised to the respective sham block data on that day. GPi Globus pallidus internus, Vent frontal horn of left lateral ventricle. STN Subthalamic nucleus, M1 primary motor cortex, LEFT left hemisphere, RIGHT right hemisphere, n.s. non-significant.

mechanisms of action could give rise to the network-level changes in electrophysiology and behaviour observed in this study. Future research will aim to interrogate these potential mechanisms.

Our results differ from previous reports that GPi beta power may increase after TUS targeted to GPi electrodes directly when pulsed at 5Hz[40]. Whilst these previously published results are of interest, we found that direct targeting of the DBS electrode causes mechanically induced artefacts at the TUS PRF, and its harmonics, increasing beta power in an agar phantom (Fig. 4). This is of particular relevance for 'on-line' TUS results where the PRF and its harmonics overlap with physiological signals of interest. In contrast, our 130 Hz pulsed TUS paradigm, targeted to connected network nodes, circumvents the risk of mechanical artefacts by both avoiding the electrode and avoiding the physiological frequency range of interest.

It is notable that both the duration of effect, and the degree of beta-suppression, appear to be heterogeneous across the four participants in this study. At the group level, we did not find a reduction in UPDRS-III which may, in part, be a result of clinical assessment timing. Since UPDRS-III assessments could not be completed during TUS ('on-line'), improvement in UPDRS-III is dependent on the duration of any

sustained effect after TUS is applied ('off-line'). Previous imaging studies provide evidence of sustained 'off-line' TUS effects however these are site specific[41], parameter dependent[42], and dynamic over time[43]. This complexity is a key challenge for the TUS community, particularly in the context of future therapeutic TUS applications. In this proof-of-concept work, we have not obtained sufficient 'off-line' electrophysiological data to quantify the duration of beta power suppression but this is likely to vary across participants given the heterogeneous nature of PD[44]. We observe, for example, that the absolute reduction in beta power in our third participant (Fig. 3A, third panel from the left) is visibly lower than that of the other three participants in this study. Post-hoc analysis of accelerometery and head tracker data revealed that this participant had significantly greater tremor amplitudes than the other participants in this study (Fig. S6). Through a combination of impaired acoustic coupling and reduced target accuracy[45], this additional head movement likely reduced the total energy delivered and may account for some of the heterogeneity observed between participants in both the magnitude and duration of effect.

Parallels can be drawn between the beta suppression reported in the DBS literature and the beta suppression observed here. Whilst the

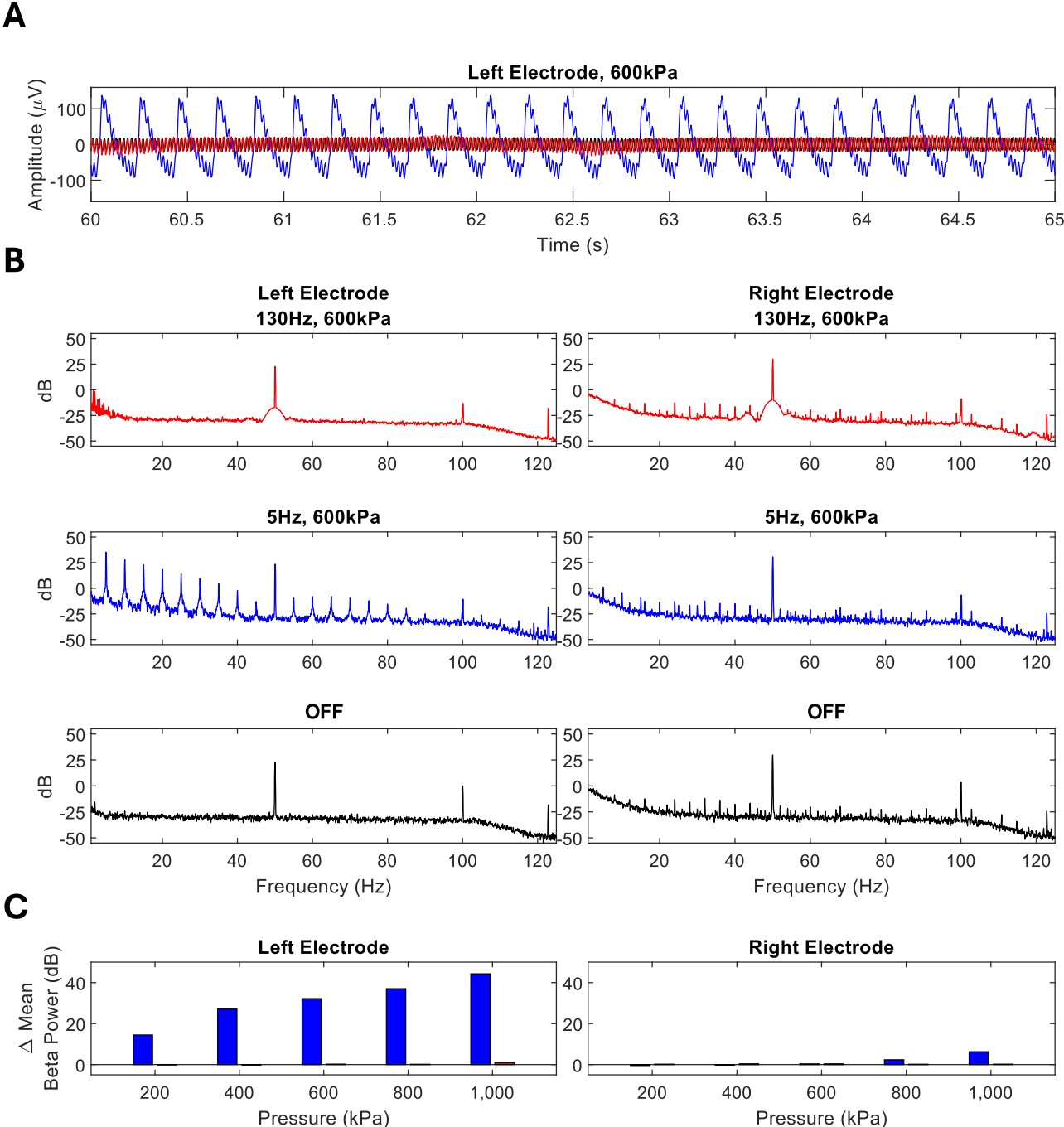

**Fig. 4 | Ultrasound induced DBS artefact characterization. A Representative data in the time domain over a 5 second period.** *Red = 130 Hz PRF, blue = 5 Hz PRF.* **B Power spectrum plots for the left and right electrodes.** *Line colours correspond to 5 Hz PRF at 600 kPa (blue), 130 Hz PRF at 600 kPa (red) and 0 kPa (black).* **C Change in Beta Power:** The change in beta power, as measured by the left and right electrodes for 5 Hz pulse repetition frequency (PRF) theta stimulation (500 kHz fundamental frequency, 20 ms pulse duration, 5 Hz PRF) and 130 Hz PRF stimulation (500 kHz fundamental frequency, 90 μs pulse duration, 130 Hz PRF). Beta power was normalised to baseline recordings, taken with the amplifier turned off. Bar colours correspond to 5 Hz PRF at 600 kPa (blue) and 130 Hz PRF at 600 kPa (red).

precise 'off-line' duration of effect was not recorded in this study, the observed duration of STN beta-suppression after STN-DBS can last up to 70 seconds, which may explain the absence of 'off-line' effects with TUS[46]. Previous evidence suggests that STN-DBS associated peak beta power reduction can range from 2.5 to 59.5%, depending on the amplitude of stimulation used, which aligns with our results[47]. It is important to note that we applied TUS to the GPi whilst recording from the STN. GPi-DBS is known to suppress GPi beta power without suppression of M1 beta power at the group level, as seen in our study[48].

However, given the invasive nature of DBS surgery and the efficacy of single-target implantation, dual-target implantation of both the GPi and the STN is rare. In a pilot study of three participants who underwent bilateral dual-target DBS, data available from two participants suggests that GPi-DBS suppresses STN beta power by up to approximately 60%[49]. Taken together, our observed results indicate that the physiological effects of TUS are similar to those of DBS.

The small sample size ($n = 4$) limits the number of comparisons and conclusions that can be made from our data. In particular,

potential drivers of variability such as levodopa equivalent daily doses and basal ganglia structural connectivity could not be explored here. Another limitation is that UPDRS-III assessments were completed between 6 and 17 minutes after each TUS block. Since the majority of PD symptoms return within the first 30 minutes after DBS discontinuation[36], differences in UPDRS-III assessment timings could contribute to score variability. Furthermore, LFP and EEG data were not recorded during these assessments so it is not known if the beta suppression observed during TUS was present during UPDRS-III. Additionally, data were not collected during STN-DBS or after dopamine administration due to technical limitations prohibiting simultaneous LFP recording during DBS and the time burden on our patient participants. Finally, this study does not investigate the effect of alternative parameter sets on beta power or bradykinesia, although previous work suggests that alternative PRFs may have opposing effects to those observed in this study[40]. This work does have a number of strengths including the randomised controlled study design using an active TUS control site; since all comparisons presented are baseline (sham) normalised GPi-TUS vs baseline (sham) normalised Ventricle-TUS, any potential fluctuations in baseline beta power are both recorded and accounted for which reduces the risk of confounds associated with these known fluctuations[20]. Additional strengths include the double blinding to condition and the use of personalised acoustic lenses which allow standardisation of peak pressures and bespoke participant-specific steering of the ultrasound focus in all three axes (medial-lateral/anterior-posterior/superior-inferior)[50,51]. Finally, post-hoc power calculations demonstrate that despite the small sample size, this study is adequately powered to detect a true difference of the magnitude observed (Fig. S7).

These results open up several possible directions for therapeutic TUS. First, exploration of TUS parameters including pulse duration and PRF may result in further enhancement in response. Second, we show that connectome-based targeting of the basal ganglia is feasible; this may improve the therapeutic success rate of TUS if results can be expected to follow those of DBS[52,53]. Third, given the above evidence that both pathological biomarkers and behaviour can be disrupted by TUS, TUS could become a tool for personalised DBS surgery. In particular, TUS should be explored for target identification and patient selection to optimise surgical success rates, particularly where the optimal target is not known.

In summary, our results show that DBS-inspired 130 Hz pulsed TUS can disrupt pathological beta oscillations in network nodes within the basal ganglia and motor cortex. This has direct implications for therapeutic translation of TUS in the context of PD and beyond. Our future work will focus on exploring drivers of variability in order to optimise TUS effects.

## Methods

This study was sponsored by the University of Oxford and conducted after NHS Research Ethics Committee approval (Reference: 23/EM/0165) and registration (trial registration no. NCT06932185), in accordance with the Declaration of Helsinki. The results presented here are from the first four participants recruited to this study. The experimental design is as described in the main text and outlined in Fig. 1C.

### Participants

Four consecutive patients were recruited from DBS follow-up clinics (Table 1) and provided informed consent to participate. Patients were eligible for participation if they had PD, DBS targeted to the STN, had a Percept® implanted pulse generator (IPG), and had a measurable beta peak on screening. Potential participants were excluded if they could not tolerate having their DBS turned off for the duration of the experiment, had an extreme language barrier and/or had a neurological/psychiatric comorbidity that may confound the results. Patients

were recruited once they were on stable dopaminergic medications and had stable DBS settings. These participants are part of an ongoing study so Data Safety and Monitoring Board (DSMB) approval was gained prior to the publication of this work.

### Surgery

The DBS surgery was performed as part of routine clinical practice 3–6 months prior to recruitment of participants into the TUS study. DBS surgery was performed under general anaesthesia. A 2.7 mm twist drill craniotomy was made and the electrode lead inserted bilaterally into the STN. All patients had intra-operative imaging to confirm electrode positioning. All electrodes were confirmed using fused pre-operative structural MRI fused with post-operative CT. Target selection was based on anatomical/stereotactic references, for electrode reconstruction in standard space, see Fig. 1D. All participants received a Percept® IPG with B33015 electrodes (Medtronic Inc., Minneapolis, Minnesota, USA).

After implantation, the DBS was programmed by a trained clinician to optimize symptom suppression and control side effects. After surgery, the stimulator was turned on to ascertain any immediate effects of stimulation on symptoms and then switched off again. The stimulator was switched on following all wound healing, one month following the operation. DBS was delivered at 125 or 130 Hz and the current was adjusted over subsequent visits to a maximum which was titrated to the maximum tolerated each patient. See Table 1 for the final stimulator settings.

### Image acquisition

Pre-operative MRI was performed on a 3T SIGNA™ Premier scanner (General Electric Healthcare, Wisconsin, USA). Structural T1-weighted MPRAGE sequences were acquired (repetition time (TR) = 2659 ms, echo time (TE) = 2.93 ms, inversion time = 1058 ms, flip angle (FA) = 8°, field of view = 256 x 256 mm, voxel size = 1 ×1 x 1 mm³) in addition to diffusion weighted sequences. The multiband diffusion weighted sequence achieved a voxel size of 1.8 × 1.8 × 1.8mm³. Diffusion weighting was applied along 109 non-colinear gradient directions ($b$ = 500 s/mm² for 15 volumes, 1000 s/mm² for 30 volumes, 2600 s/mm² for 64 volumes), with five non-diffusion ($b$ = 0) weighted volumes (TR = 6233 ms, TE = 73.9 ms, FA = 90°). An additional $b$ = 0 image was acquired with an opposing phase encoding direction was also acquired for distortion correction. Post-operative CT scans were taken as mentioned above for verification of electrode position (Siemens Naeotom Alpha Version VA50, Siemens Healthineers, Forchheim, Germany). A standard CT brain protocol was used (scan mode: routine spiral adult head quantum plus Xcare; tube voltage: 120 kV; effective mAs 311; acquisition 96 × 0.4 mm volume CT dose index (CTDIvol): 55.7; pitch: 0.35; rotation time (seconds) 0.5; reconstruction slice thickness: 0.6 mm; slice increment: 0.4 mm; reconstruction kernel: HR44; quantum iterative reconstruction [QIR] strength: 3; matrix size: 512 × 512).

### Image preprocessing

Image pre-processing was carried out using the FMRIB Software Library (FSL; Oxford, UK). Susceptibility-induced off-resonance field was estimated using *topup* using $b$ = 0 volumes with opposing PE directions. *Eddy* was then used to correct for motion and eddy currents. *BEDPOSTX* was used for ball and stick modelling of local diffusion parameters, with up to three crossing fibres per voxel. Pre-operative T1-weighted MRI images were registered to post-operative CT images using *FLIRT* and registered to Montreal Neurological Institute (MNI) standard space using *FNIRT*. The inverse warp was then applied to the left STN and left GPi masks in the DISTAL atlas[54], and these were then binarised to produce left STN and left GPi masks in native space.

## Tractography based targeting

Probabilistic tractography was carried out using *PROBTRACX* with modified Euler streaming with the native STN mask set as the seed and the GPi set as a waypoint and termination mask. The output (fdt map) was then transformed to CT space and the 'peak connectivity' region between in the GPi was set as the single voxel target for onward ultrasound point source simulation. This target was visualised in native T1-weighted imaging to verify target location. An important note is that this 'peak connectivity' region represents the highest probability of tract existing between the STN and GPi.

## Personalised acoustic lens design

Two personalised acoustic lenses were designed for each participant, one targeting the peak connectivity voxel in the GPi and another targeting the frontal horn of the left lateral ventricle. A similar design process is described elsewhere[15]. In brief, each participant's CT was segmented to bone, soft-tissue/cerebrospinal fluid (CSF) and electrode and assigned acoustic or thermal properties given in Table S1. Each target was then set for a 'backward' simulation, where a point source was placed at the target and the acoustic wave propagated out to the receiver/sensor plane. The amplitude map was used to determine the ultrasound transducer placement by maximizing the amplitude over the surface at the transducer. The phase was used to design a phase-conjugate lens mounted to the transducer to account for the aberration due to the propagation path was then carried out using ultrasound transducer and acoustic lens to determine the acoustic field inside the head (Fig. 3A). The ultrasound focus was then verified in the structural MRI with the co-registered CT to ensure that the ultrasound focus overlapped with the tractography 'peak connectivity' region, and that the electrode was avoided.

Numerical simulations of the acoustic wave equation were performed using the GPU-accelerated k-Wave Toolbox[55] code on a Quadro RTX 6000 graphics card (NVIDIA, Santa Clara, CA, USA). Simulations were performed at a spatial resolution of 0.5 mm in all three dimensions. The Courant-Friedrichs-Lewy number was set to 0.28 based on the highest sound speed in the domain.

## Thermal simulations

Thermal simulations of the Pennes' bioheat equation were performed using the k-Wave Toolbox to ensure that temperature safety limits were met. For each participant the same grid and segmentation was used as in the ultrasound simulations with the thermal properties given in the supplementary information. For each lens the heat source was derived from the predicted acoustic field map and pulsing protocol (Fig. 3B) and the temperature simulated over the entirety of the experiment. Acoustic lenses were only manufactured after confirmation that temperature does not rise above 2 ° C anywhere in the head. Thermal simulations were validated in a porcine ex-vivo model (Fig. S4). Tissue samples were acquired from a local butcher and not subject to ethical approval. The handling and disposing of samples followed standard University of Oxford guidelines for the use of animal-derived tissue in a laboratory setting.

## Acoustic lens fabrication

Lens positives were 3D printed using a Formlabs Form 3B resin 3D printer (Clear v4 resin, 0.1 mm layer height). After curing, lens positives were used to make silicone (Dragon Skin 10 SLOW, Smooth-On), negative moulds. Negative moulds were filled with SYLGARD 184 silicone elastomer polydimethylsiloxane (PDMS), degassed in a centrifuge at 3000 rpm for 1 minute and placed in a near vacuum for 30 minutes. To avoid the bonding of the silicone negative mould to the PDMS lens, negative moulds were plasma treated, deposited with polyvinyl alcohol and sprayed with the a spray release agent (Ease Release 200, Smooth-On). PDMS lenses were cured in a 60 °C oven for 12 hrs.

## Acoustic lens validation

Each lens was visually inspected for abnormalities and validated in degassed water using a needle hydrophone (model HNA-400, ONDA, Sunnydale, CA, USA). Each lens and silicone membrane were mounted to the transducer under degassed water as illustrated in supplementary Fig. 1. 2D pressure maps were measured around the peak pressure region with a fixed axial distance (field of view = 40 mm x 40 mm, spatial resolution = 1 mm). Pressure readings were taken as the peak rarefaction pressure averaged over 10 pulses (500 kHz, 20 cycles per pulse, 10 ms pulse period). Resultant pressure maps were accepted or rejected based on a visual inspection. All lenses were retested overnight before each experimental session with lenses placed in their final mounted position. Simulated and measured free field acoustic pressure fields for all lenses can be found in the Supplementary Information Fig. 2.

## Ultrasound stimulation

Sonications were performed using a single element planar US transducer (H294, Sonic Concepts Inc, Bothwell, WA, USA) 64 mm diameter, with a 500 kHz fundamental frequency fitted with a personalised lens bespoke to each target. Each acoustic lens was mounted to the transducer while immersed in degassed water, a silicone membrane was attached and filled with degassed water. During the experiment, ultrasound gel was combed into the participants hair and applied to the silicone membrane.

The target order was randomised and each target was sonicated on separate consecutive days. On each day, a sham TUS block preceded an active TUS block and followed the same format (Fig. 3C). Simultaneous online neuronavigation was performed using with Brainsight v2.5.5 (Rogue Research Inc., Montréal, Québec, Canada) using the T1-weighted MR images from each participant.

In each of the five, 5 minutes, active TUS blocks, ultrasound was delivered in 800 ms pulse trains repeated every 2.5 seconds. Within each pulse train, individual ultrasound pulses had a 90µs pulse duration and a 130 Hz pulse repetition frequency (PRF) corresponding to a duty cycle of 1.17% (Fig. 1B). The ultrasound stimulation waveform was digitally generated using Python 3.8.10 at a sampling rate of 10 MHz and converted to an analogue voltage with an arbitrary waveform generator (Handyscope HS5, TiePie, WL Sneek, The Netherlands). This voltage was amplified with a 55 dB broadband amplifier (1040 L, E&I, Rochester, NY, USA) and connected to the transducer through a 50Ω electrical impedance matching network.

We have previously shown that pulsed TUS results in an auditory artefact that can be masked by playing an audible signal to participants through earphones[21]. Here a 130 Hz square-wave auditory masking signal was played through earphones during all blocks (active and sham). This signal was created using the simpleaudio package for Python 3.8.10.

## Electrophysiology data recording

LFPs were recorded wirelessly using the Medtronic Percept's 'Indefinite Streaming' mode which streamed to the tablet programmer. Bilateral STN-LFPs were recorded (bipolar left 0–2, 1–3, bipolar right 0–2, 1–3) at a sampling rate of 250 Hz and the data recorded from the left and right hemisphere were considered independently. Electroencephalography (EEG) was recorded using a Saga amplifier (TMS International) at a sampling rate of 4096 Hz. Individual electrodes were placed at FP1, FP2, F3, Fz, F4, C3, Cz, C4, P3, Pz and P4 with Ten20 conductive paste. Since we delivered TUS through the temporal window, these midline electrodes were selected to ensure that our EEG electrodes would not interact with our ultrasound field. Electrodes were located using the anatomical landmarks outlined in the 10-20 system for localizing EEG electrodes.

Electrophysiological data recorded from the DBS electrodes was time locked to the task and EEG using transcutaneous electrical nerve

stimulation (TENS). Electrical pulses were generated using a biphasic constant current stimulator (DS8R, Digitimer, Hertfordshire, UK). Subthreshold amplitudes was set for each participant to the maximum value that they could not perceive by ramping up the current from 0.1 mA in 0.1 mA increments until they reported a sensation. After which the current was reduced until they no longer reported this sensation.

### Electrophysiology preprocessing

LFP data were then extracted via a tablet programmer and stored in a laboratory computer for offline processing. The Perceive Toolbox (https://github.com/neuromodulation/perceive) was used to convert the stored JSON files to MAT-files for further analysis in MATLAB (MathWorks, Natick, MA). For the recorded EEGs, bipolar signals were constructed offline by differentiating between 'C3' and 'Cz' (i.e. 'C3Cz') or 'C4' and 'Cz' (i.e. 'C4Cz'). Only C3Cz and C4Cz bipolar channels were analysed here. EEG was time-locked and down-sampled to a sampling rate of 250 Hz to match the LFP sampling rate. LFP and EEG data were then band-pass filtered between 4-45 Hz, detrended and DC removed.

### Spectral analysis

After preprocessing, resting power spectral density (PSD) was estimated using Welch's overlapped segment averaging estimator for the LFP, EEG, accelerometry and head tracker recordings of each individual for the entirety of each rest block (approx. 5 minutes) using *pwelch*, with a Hann window of 1 second and a window overlap of 600 ms. The PSD was then broadband total power normalised. The difference between the PSD of each TUS and sham block was taken in order to account for fluctuations in beta power across days. The relative beta power (13–30 Hz) difference between GPI-TUS and Ventricle-TUS was then computed for each participant in both the LFP and EEG data.

### Coherence

Coherence (magnitude-squared coherence) was calculated using Welch's averaged modified periodogram method with a 1 second window and a frequency resolution of 1 Hz using *mscohere*. Coherence was analysed from 4 to 45 Hz and difference in coherence from baseline was computed and presented.

### Behavioural tasks

Participants practiced each task the day before the experiment under guided supervision until they clearly demonstrated task comprehension. In addition, instructions were displayed before each block throughout the experiment. Behavioural tasks were presented on a 1600 × 900 LCD display with participants sat at a comfortable distance. All tasks were created using PyschoPy 2024.2.4 with Python 3.8.10. A photodiode was affixed to the top of the display with opaque black tape to time lock the EEG data to the presentation of new stimuli.

Throughout the experiment participants were encouraged to remain stationary and attend to the display. As shown in Fig. 3C. all tasks were initially presented in a sham block without the application of TUS. An auditory mask was played through earphones throughout, and the participant was blinded throughout. Participants were asked to partake in a series of tasks: rest, stop-signal task and random-dot motion task. A break of 5 minutes was present between each block to allow participants to relax and to mitigate any potential heating effects from TUS.

Each block started with an initial rest period. During this time a white fixation cross was presented on a black background in the centre of the screen for 5 minutes which participants were instructed to attend to while remaining stationary.

After a 5 minute break, participants were presented the stop-signal task (SST). Participants were instructed to click the left or right arrow key in accordance with the direction of the arrows presented on screen. 24 go trials were presented at random time intervals between

2.6 and 3.4 s, the median reaction time to these stimuli were recorded. After this, participants were presented with instructions informing them that in subsequent trials they must withhold their response if a red cross appeared. 78 go trials, and 26 no-go trials were presented, if a participant responded more than 100 ms slower than their median reaction time during the initial go trials a "Too slow" message appeared. The stop signal delay (SSD), the time delay before the presentation of the stop signal (red cross) was adjusted on a trial-by-trial basis to maintain a stop success of 50%. SST data has not yet been analyzed due to omission rates and violations of the race model that cannot be accommodated in this small sample size[56].

Participants were then presented with the random-dot motion (RDM) task which was used to assess rection times. Prior to each block participants were instructed to respond to the coherent movement of dots by clicking the corresponding left or right mount buttons as soon as they knew the answer. At random time intervals between 1.25 and 1.75 seconds dots moved coherently in the left or right direction. These coherent movements included either 8% of dots (low coherence) or 50% of dots (high coherence). Reaction times between 0.1 and 2 seconds from stimulus onset were recorded. The RDM task was repeated twice in approximate 5 minute blocks with a 5 minute break between each task. Reaction times were min/max normalised for all 240 trials completed by that participant on any day. Median reaction time of each block was taken for onward analysis. Data from each for each active TUS block was normalised to the sham block on the same day.

### Clinical measures

An abbreviated Unified Parkinson's Disease Rating Scale part III (UPDRS-III) assessment was completed with each participant at baseline each morning and at the end of each sham and active block. Given that participants were seated and connected to our EEG amplifier for all assessments, only a subset of the UPDRS could be recorded. Walking and pull-back tests were not completed. Assessments were recorded and scored offline by a blinded assessor. Rigidity was assessed during each UPDRS assessment by a blinded assessor. It was not possible to deliver TUS during UPDRS-III assessments so these assessments can be considered 'off-line', i.e. after TUS had stopped.

### Statistics & reproducibility

Statistical analysis was completed in MATLAB (MathWorks, Natick, MA). Owing to the small sample size, only a limited number of comparisons were made and these were defined a priori. Lillifors test was used to test for normality and paired t-tests were used and correlations were tested accordingly. Adjustment for multiple comparisons was completed using the false discovery rate approach described by Benjamini and Hochberg[57]. Note that both the participant and the assessor was blinded to condition during the study.

### Reporting summary

Further information on research design is available in the Nature Portfolio Reporting Summary linked to this article.

## Data availability

Source data are provided with this paper. Raw, anonymised data are available upon request from the corresponding author on final publication of the ongoing study. Source data are provided with this paper.

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

## Acknowledgements

First and foremost we would like to acknowledge the participants who agreed to take part in this study, altruistically giving significant time and energy to potentially help others in the future through improved understanding of transcranial focused ultrasound. This study is funded by the Rosetrees Trust and the John Black Charitable Trust (Ref: ID2021\100054). JE is funded through a Medical Research Council Clinical Research Training Fellowship (Ref: MR/X006417/1). TD is funded by the Royal Academy of Engineering. S.H. was supported by the Guarantors of Brain, The Royal Society (IES\R3\213123), and Parkinson's UK. Additional support was provided by the NIHR Oxford Health Biomedical Research Centre (NIHR203316). The views expressed are those of the authors and not necessarily those of the NIHR or the Department of Health and Social Care. We would like to acknowledge Professor Elsa Fouragnan University of Plymouth, who provided invaluable training on using the Brainsight neuronavigation system for our experiments. We would like to acknowledge Dr James Grist, Oxford Centre for Clinical Magnetic Resonance Research, University of Oxford, in addition to the team of radiographers there, who provide research-grade clinical scans for the Functional Neurosurgery service. We would like to acknowledge our movement disorder nursing team, particularly Beth Petric, Claie Fletcher and Laura Bacchini for ongoing help and support. We would like to acknowledge Dr Damian Herz, University Hospital Heidelberg, who provided a copy of his random dot-motion task and Professor Huiling Tan, University of Oxford, who provided access to laboratory space and equipment. We would also like to acknowledge Moaad Benjabar for his support in device benchtop testing.

## Author contributions

R.C., A.G., C.B., T.D. and J.E. were involved in conceptualization. R.C., A.G., C.B., T.D., X.C. and J.E. were involved in the development of methodology. J.E., J.T., J.H., S.H., M.E.S., A.Z. and J.F. were involved in the investigation. J.E. and J.T. were contributed to visualization. R.C., A.G., T.D. and J.E. acquired funding. R.C., A.G. and J.E. contributed to study administration. R.C. and A.G. were responsible for supervision. The original manuscript draft was written by J.E. and J.T. and all authors contributed in the review and editing of the final manuscript.

## Competing interests

The authors declare no competing interests.
