## [Transparent Peer Review file · Nature Communications]

Suppression of Pathological Oscillations with Transcranial Focused Ultrasound in Parkinson's Disease

Corresponding Author: Dr John Eraifej

Version 0:

Reviewer comments:

Reviewer #1

(Remarks to the Author)

This randomized controlled cross-over study presents the first evidence that transcranial ultrasound stimulation (TUS) pulsed at 130Hz can non-invasively suppress pathological beta-band oscillations in the subthalamic nucleus (STN) and improve bradykinesia in patients with Parkinson's Disease (PD). By recruiting four PD patients with implanted sensing-enabled DBS electrodes, the authors successfully demonstrated that GPi-TUS reduces ipsilateral STN beta power by approximately 10.34% and reaction time by 17.70%.

While the study provides groundbreaking evidence for the modulation of deep brain biomarkers, the sample size of four participants is a significant limitation for generalizability. Although the authors acknowledge this limitation, it is recommended to explicitly frame the manuscript as a "pilot study" or "proof-of-concept" investigation. If possible, including a post-hoc power analysis would help validate the statistical strength of the observed effects despite the small N.

The results show a significant improvement in reaction time (17.70%) during the "online" task, yet there was no significant reduction in the UPDRS-III scores at the group level. This discrepancy requires a deeper discussion regarding the "washout" period of TUS effects. If the therapeutic benefits are strictly limited to the stimulation period (online effects), the discussion should address the implications for TUS as a clinical therapy and potential strategies to induce neuroplasticity or longer-lasting (offline) effects.

The study demonstrates that 130Hz TUS mimics the effects of 130Hz electrical DBS. While the "indirect pathway" hypothesis is mentioned, the manuscript would benefit from a more detailed theoretical discussion on how the mechanical effects of ultrasound at this specific pulse repetition frequency (PRF) translate to neuronal modulation (inhibition vs. excitation) in the GPi, as opposed to the electrical mechanisms of DBS.

Reviewer #2

(Remarks to the Author)

This manuscript investigates the impact of low-frequency transcranial ultrasound stimulation (TUS) targeting the GPi in Parkinson's disease patients with implanted STN DBS leads. By recording STN LFP activity intraoperatively during active and sham sonication conditions, the authors demonstrate that GPi-TUS can transiently suppress pathological beta oscillations in the ipsilateral STN.

Overall this is a very interesting study that fills a major gap in the low frequency FUS field, in which the stimulation parameters and the effects on the target structures are unknown. This study provides one of the first evidence that low frequency FUS alters pathological network oscillations which has similar effects as high frequency DBS. The strengths of the study include: multisite multimodal network perturbation and recording to uncover potential mechanisms of FUS; personalized acoustic lens to maximize accuracy and focality of proposed stimulation field as demonstrated in supplementary figures, well-characterized ultrasound-induced artifacts on DBS electrodes.

However, a number of concerns should be addressed to strengthen the conclusions and generalizability of the study:

- The study has a very small sample size and one of the study subjects (3rd panel in figure 3A) did not show much, if any,

beta suppressions with GPI-TUS compared to the other sham conditions. Can the authors elaborate on why they think this subject did not show corresponding beta STN suppression?

- Does GPI-TUS suppress STN beta beyond the duration of TUS? There are several studies and lines of evidence that GPI stimulation can trigger neuroplasticity and effects of stimulation can last beyond cessation of stimulation.
- What is the duration for each condition (sham/active GPI-TUS or ventricle TUS). How long of a resting period was there between blocks?
- Did patients take their dopamine medication during the testing or in the middle of a block? Can the authors comment on the effect of medication on their STN beta if they did take medicine in between the blocks.
- Was sham TUS always applied before active TUS? The order of GPI vs ventricle is randomized, but if the order of sham vs active is not counterbalanced or randomized, this could confound the results.
- What is the effect of TUS on M1-STN coherence?
- The authors picked 130 Hz PRF to mimic the frequency of DBS stimulation, have they used another PRF (such as 30 Hz or 60 Hz), where it should have minimal effect on STN beta as a control?

Reviewer #3

(Remarks to the Author)

The manuscript provides with a novel application of TUS, demonstrating the possibility of obtaining similar results to DBS for PD through a non-invasive technique, which seems highly relevant to the field. The authors also demonstrate that the same biomarkers used for DBS can be used under a TUS procedure. The authors provide a clear and concise description of the protocol and analysis followed. Although some relevant information regarding the experiment setup is missing from the main manuscript, it is included in the supplementary materials.

However, the manuscript would benefit from the following suggestions to improve its clarity and further validate some of the claims from the study.

Major points:

- Always have the sham session before the active session could lead to a training effect improving outcomes in the active session. The authors might want to clarify that they avoided this problem by normalising all their results with the active ventricle (rather than with the sham).
- For the change in reaction time, to what extent have outliers influenced the results? In other words, what is the reaction time difference observed for the mean (compared to the median)?
- In this manuscript, the presented DBS-inspired TUS protocol can modulate disease-related pathological oscillations in the same direction as known therapies such as DBS for PD. The same biomarker (beta-band power) as for DBS is analysed and a reduction in beta-band power is reported. However, to further validate that the proposed protocol can provide similar outcomes as established DBS protocols, it would be good to include some comparison between the results obtained in the study and the results (e.i., common beta-band power reduction) obtained in established DBS studies. Is the beta-band power reduction obtained from TUS comparable to the beta-band power reduction commonly obtained with DBS?
- In the discussion, using acoustic lenses is mentioned as an advantage of the current study. Could the authors discuss how lenses enabled the current study? What would have happened without lenses (e.g. stimulating adjacent tissue)?

Minor points:

- p. 6: (right -> (right)
- Fig. 3A: it is difficult to see differences at the moment. Maybe the colour scheme could be changed so that active is in dark colours and sham in lighter shades of the same colours (e.g. GPI in dark/light red and Ventricle in dark/light blue).
- The placement of Figures 1 and 2, and Table 1, could be improved to make the manuscript clearer. It is recommended to locate the Figures and Tables in the order that they are referred to in the manuscript (e.g., Table 1 is referred to first, so it should be located before Figure 1 and 2 instead of after).
- On the supplementary materials, for easier reproducibility of the study, it would be good to specify if the EEG electrodes used are water or gel based, since both options seem to be available for the described amplifier.
- Fig. S3B is too small to be read correctly. Please, increase the size of the individual plots.
- On the supplementary materials, it would be good to describe how were the EEG electrodes located. Did you make use of a commercially available EEG cap as is common with the described amplifier or did you place the electrodes directly on the patients' head? Did the placement of the electrodes interfere in any case with the placement of the transducer?

Reviewer #4

(Remarks to the Author)

Version 1:

Reviewer comments:

Reviewer #1

(Remarks to the Author)

I am satisfied with the authors' responses. Thanks.

Reviewer #2

(Remarks to the Author)

The authors have adequately addressed my questions and provided clarification and details that have strengthened their manuscript.

Reviewer #3

(Remarks to the Author)

The authors have successfully addressed all points.

Reviewer #4

(Remarks to the Author)

Reviewer #1 (Remarks to the Author):

This randomized controlled cross-over study presents the first evidence that transcranial ultrasound stimulation (TUS) pulsed at 130Hz can non-invasively suppress pathological beta-band oscillations in the subthalamic nucleus (STN) and improve bradykinesia in patients with Parkinson's Disease (PD). By recruiting four PD patients with implanted sensing-enabled DBS electrodes, the authors successfully demonstrated that GPI-TUS reduces ipsilateral STN beta power by approximately 10.34% and reaction time by 17.70%.

While the study provides groundbreaking evidence for the modulation of deep brain biomarkers, the sample size of four participants is a significant limitation for generalizability. Although the authors acknowledge this limitation, it is recommended to explicitly frame the manuscript as a "pilot study" or "proof-of-concept" investigation. If possible, including a post-hoc power analysis would help validate the statistical strength of the observed effects despite the small N.

We thank the reviewer for their kind words. We agree that this work represents a "proof-of-concept", demonstrating that biomarkers of disease *can* be modulated by TUS in a small number of patients with Parkinson's disease. We have therefore reframed the study as a "proof-of concept" study in the abstract, main text and discussion of this manuscript.

'In this randomised controlled proof-of-concept study, we leverage PD as a model system and show, for the first time that DBS-inspired 130Hz pulsed GPI-TUS reduces STN beta-band power.'

'In this proof-of-concept work, we have not obtained sufficient 'off-line' electrophysiological data to quantify the duration of beta power suppression but this is likely to vary across participants given the heterogeneous nature of PD⁴⁵'

Page 2 (38), Page 3 (69), Page 10 (251)

As suggested, we have also completed a post-hoc power analysis. This analysis suggests that a sample size of 4 would provide a power of ≥ 0.9 for both suppression of STN beta and reduction reaction time during a behavioural task. This post-hoc power analysis has been included in the supplementary information provided in this manuscript (Figure S5).

'Finally, post-hoc power calculations demonstrate that despite the small sample size, this study is adequately powered to detect a true difference of the magnitude observed (Fig. S7)'

Page 11 (291-293), Supplementary information, figure S7, page 17 (333-334)

The results show a significant improvement in reaction time (17.70%) during the "online" task, yet there was no significant reduction in the UPDRS-III scores at the group level. This discrepancy requires a deeper discussion regarding the "washout" period of TUS effects. If the therapeutic benefits are strictly limited to the stimulation period (online effects), the discussion should address the implications for TUS as a clinical therapy and potential strategies to induce neuroplasticity or longer-lasting (offline) effects.

We thank the reviewer for this suggestion and agree that this is a very important question for the field.

To address this we have included an additional paragraph in the discussion which discusses this in some detail, with a focus on possible explanations for our observations. In this paragraph we also discuss the heterogeneity of response observed in our participants and its potential relevance to the heterogeneity in UPDRS-III scores (which were assessed 'off-line') as this also answers a question raised by reviewer 2.

Page 10 (244-260)

The study demonstrates that 130Hz TUS mimics the effects of 130Hz electrical DBS. While the "indirect pathway" hypothesis is mentioned, the manuscript would benefit from a more detailed theoretical discussion on how the mechanical effects of ultrasound at this specific pulse repetition frequency (PRF) translate to neuronal modulation (inhibition vs. excitation) in the GPi, as opposed to the electrical mechanisms of DBS.

We thank the reviewer for this suggestion, we have now included a more detailed explanation in the discussion section.

Page 9 (207-235)

Reviewer #2 (Remarks to the Author):

This manuscript investigates the impact of low-frequency transcranial ultrasound stimulation (TUS) targeting the GPi in Parkinson's disease patients with implanted STN DBS leads. By recording STN LFP activity intraoperatively during active and sham sonication conditions, the authors demonstrate that GPi-TUS can transiently suppress pathological beta oscillations in the ipsilateral STN.

Overall this is a very interesting study that fills a major gap in the low frequency FUS field, in which the stimulation parameters and the effects on the target structures are unknown. This study provides one of the first evidence that low frequency FUS alters pathological network oscillations which has similar effects as high frequency DBS.

The strengths of the study include: multisite multimodal network perturbation and recording to uncover potential mechanisms of FUS; personalized acoustic lens to maximize accuracy and focality of proposed stimulation field as demonstrated in supplementary figures, well-characterized ultrasound-induced artifacts on DBS electrodes.

We thank the reviewer for this kind summary.

However, a number of concerns should be addressed to strengthen the conclusions and generalizability of the study:

- The study has a very small sample size and one of the study subjects (3rd panel in figure 3A) did not show much, if any, beta suppressions with GPi-TUS compared to the other sham conditions. Can the authors elaborate on why they think this subject did not show corresponding beta STN suppression?

We thank the reviewer for this astute observation. Anecdotally, during the experiments we noted that this participant had a very severe tremor, unlike the other participants in this study.

The reviewer's comment prompted us to look at this more closely; we have now quantified tremor magnitude in both the accelerometry data and the head tracker data available. The power spectral density of both analyses demonstrate that there is a clear 4-5Hz resting tremor in the participant's right arm (as measured with an accelerometer) and that this translates to a head tremor (measured as a rotational movement; see Fig S6).

We think there are two possible consequences of this. Firstly, the participant's movement may have meant that participant 3 received less TUS to target. Secondly, a single large movement can uncouple the ultrasound transducer from the scalp – the likelihood of this occurring increases significantly with increased movement. Once uncoupled from the scalp, TUS will no longer be delivered to target.

We have now highlighted this in the manuscript in the discussion section discussing heterogeneity of response.

'We observe, for example, that the absolute reduction in beta power in our third participant (Fig 3A, third panel from the left) is visibly lower than that of the other three participants in this study. Post-hoc analysis of accelerometry and head tracker data revealed that this participant had significantly greater tremor amplitudes than the other participants in this study (Fig. S6).'

Page 10 (253-257), Supplementary information, figure S6, page 16 (319-321)

- Does GPi-TUS suppress STN beta beyond the duration of TUS? There are several studies and lines of evidence that GPi stimulation can trigger neuroplasticity and effects of stimulation can last beyond cessation of stimulation.

We thank the reviewer for this question. It is a similar question to that raised by reviewer 1, specifically discussing the potential 'off-line' effects and the duration that they might last.

In this initial dataset we do not have off-line LFP data to confidently answer this question and recognise this as a limitation of our study.

We have therefore elaborated on this limitation in the discussion and highlighted the need for off-line data collection in future.

Page 10 (244-260)

- What is the duration for each condition (sham/active GPI-TUS or ventricle TUS). How long of a resting period was there between blocks?

Each condition is lasts 5 minutes and each rest period between blocks is also 5 minutes. This is now clarified in the Supplementary Information.

Supplementary Information, page 5 (149), page 7 (211, 216), page 8 (233)

- Did patients take their dopamine medication during the testing or in the middle of a block? Can the authors comment on the effect of medication on their STN beta if they did take medicine in between the blocks.

We thank the reviewer for this suggested clarification.

All experiments were conducted between medication doses and timings were fixed on each day so to avoid the potential confound associated with medication timings.

We have now added a sentence to clarify this in the manuscript.

'All sham and TUS experiments were completed between dopamine replacement therapy doses.'

Page 6 (114-115)

- Was sham TUS always applied before active TUS? The order of GPI vs ventricle is randomized, but if the order of sham vs active is not counterbalanced or randomized, this could confound the results.

We thank the reviewer for this comment. This was also raised as a point of clarification by reviewer 3.

It is indeed true that sham experiments are always conducted before active experiments but this is done to allow for baseline normalisation on each day. The only comparisons made are between baseline (sham) normalised ventricle data and baseline (sham) normalised PD data.

We did this in order to reduce the risk of confounds from variations in baseline beta power.

'since all comparisons presented are baseline (sham) normalised GPI-TUS vs baseline (sham) normalised Ventricle-TUS, any potential fluctuations in baseline beta power are both recorded and accounted for which reduces the risk of confounds associated with these known fluctuations²⁰.'

Page 11 (286-289)

- What is the effect of TUS on M1-STN coherence?

We thank the reviewer for this question as it has prompted us to investigate this.

We find no significant difference in M1-STN coherence which is perhaps unsurprising since we are applying TUS to an external node to this particular structural connection (the GPI). Whilst there may

be a trend towards a relative increase in high beta coherence, this does not pass statistical significance after correction.

We now have included this finding in the results, discussion and supplementary information.

'There was no significant change in M1-STN beta coherence at the group level (Fig S3).'

'bypassed by the hyperdirect pathway, a monosynaptic cortico-subthalamic projection. Similarly, we found a correlated reduction in ipsilateral M1-STN beta power without significant modulation in M1-STN coherence'

Page 6 (141-142), Page 9 (216-219), Supplementary information, figure S3, page 12 (272-276), page 6 (196-200)

- The authors picked 130 Hz PRF to mimic the frequency of DBS stimulation, have they used another PRF (such as 30 Hz or 60 Hz), where it should have minimal effect on STN beta as a control?

Many thanks for raising this very important limitation. In our study, we did not explore the parameter space as our primary research question focused on whether TUS, when applied at DBS parameter sets (130Hz PRF), would have comparable effects to electrical stimulation with implanted electrodes. To confirm these effects are site specific we used an active control (ventricle) but do not have any alternative frequencies as controls. Previous studies of TUS to the same target would indicate that alternative PRFs may increase beta power (the opposite of our finding) which we have now highlighted in the limitations section of our discussion:

'Finally, this study does not investigate the effect of alternative parameter sets on beta power or bradykinesia, although previous work suggests that alternative PRFs may have opposing effects to those observed in this study⁴¹.'

Page 11 (283-285)

Reviewer #3 (Remarks to the Author):

The manuscript provides with a novel application of TUS, demonstrating the possibility of obtaining similar results to DBS for PD through a non-invasive technique, which seems highly relevant to the field. The authors also demonstrate that the same biomarkers used for DBS can be used under a TUS procedure. The authors provide a clear and concise description of the protocol and analysis followed. Although some relevant information regarding the experiment setup is missing from the main manuscript, it is included in the supplementary materials.

We thank the reviewer for their kind words and their interpretation of our results.

However, the manuscript would benefit from the following suggestions to improve its clarity and further validate some of the claims from the study.

Major points:

- Always have the sham session before the active session could lead to a training effect improving outcomes in the active session. The authors might want to clarify that they avoided this problem by normalising all their results with the active ventricle (rather than with the sham).

We thank the reviewer for this helpful suggestion, this point of clarification was also raised by reviewer 2. This has been clarified in the manuscript

Page 11 (286-289)

- For the change in reaction time, to what extent have outliers influenced the results? In other words, what is the reaction time difference observed for the mean (compared to the median)?

We thank the reviewer for this suggested sensitivity analysis. We have now also calculated the mean difference in reaction time and find that there is a similar magnitude of reaction time reduction for the mean (18.6%) as for the median (17.7%).

Since the reaction time data are not normally distributed, we have kept the results for the median in the main body of the text but include the results for the mean in the supplementary information and describe it as a sensitivity analysis.

Supplementary information, figure S8, page 17 (326-328)

- In this manuscript, the presented DBS-inspired TUS protocol can modulate disease-related pathological oscillations in the same direction as known therapies such as DBS for PD. The same biomarker (beta-band power) as for DBS is analysed and a reduction in beta-band power is reported. However, to further validate that the proposed protocol can provide similar outcomes as established DBS protocols, it would be good to include some comparison between the results obtained in the study and the results (e.i., common beta-band power reduction) obtained in established DBS studies. Is the beta-band power reduction obtained from TUS comparable to the beta-band power reduction commonly obtained with DBS?

We thank the reviewer for this comment and agree that this comparison with published DBS literature was lacking in our discussion.

To address this we have now included a paragraph outlining the common beta-band reduction seen in the literature for both STN and GPi DBS, as well as STN beta-band suppression when applying DBS to the GPi.

Page 10 (261-273)

- In the discussion, using acoustic lenses is mentioned as an advantage of the current study. Could the authors discuss how lenses enabled the current study? What would have happened without lenses (e.g. stimulating adjacent tissue)?

We have now elaborated on this statement and referenced the evidence behind the use of acoustic lenses when compared to alternatives.

Since we used an unfocused transducer, an acoustic lens is necessary to create a TUS focus at target.

'Additional strengths include the double blinding to condition and the use of personalised acoustic lenses which allow standardisation of peak pressures and bespoke participant-specific steering of the ultrasound focus in all three axes (medial-lateral/anterior-posterior/superior-inferior)^{51,52}'

Page 11 (289-292)

Minor points:

- p. 6: (right -> (right)

Many thanks, this has been corrected.

Page 6 (137)

- Fig. 3A: it is difficult to see differences at the moment. Maybe the colour scheme could be changed so that active is in dark colours and sham in lighter shades of the same colours (e.g. GPI in dark/light red and Ventricle in dark/light blue).

Many thanks for this suggestion, we have now implemented this in the revised Fig 3A.

Page 6 (148)

- The placement of Figures 1 and 2, and Table 1, could be improved to make the manuscript clearer. It is recommended to locate the Figures and Tables in the order that they are referred to in the manuscript (e.g., Table 1 is referred to first, so it should be located before Figure 1 and 2 instead of after).

Many thanks, this has been amended.

Page 3 (72)

- On the supplementary materials, for easier reproducibility of the study, it would be good to specify if the EEG electrodes used are water or gel based, since both options seem to be available for the described amplifier.

Many thanks, this has now been clarified.

'Individual electrodes were placed at FP1, FP2, F3, Fz, F4, C3, Cz, C4, P3, Pz and P4 with Ten20 conductive paste. Since we delivered TUS through the temporal window, these midline electrodes were selected to ensure that our EEG electrodes would not interact with our ultrasound field. Electrodes were located using the anatomical landmarks outlined in the 10-20 system for localizing EEG electrodes.'

Supplementary Information, page 6 (167-171)

- Fig. S3B is too small to be read correctly. Please, increase the size of the individual plots.

Many thanks, these plots have now been increased in size with an increase in font size.

Supplementary Information, figure S4, page 13 (280)

- On the supplementary materials, it would be good to describe how were the EEG electrodes located. Do you make use of a commercially available EEG cap as is common with the described amplifier or did you place the electrodes directly on the patients' head? Did the placement of the electrodes interfere in any case with the placement of the transducer?

Many thanks, this has now been clarified.

'Individual electrodes were placed at FP1, FP2, F3, Fz, F4, C3, Cz, C4, P3, Pz and P4 with Ten20 conductive paste. Since we delivered TUS through the temporal window, these midline electrodes were selected to ensure that our EEG electrodes would not interact with our ultrasound field. Electrodes were located using the anatomical landmarks outlined in the 10-20 system for localizing EEG electrodes.'

Supplementary Information, page 6 (167-171)

Reviewer #4 (Remarks to the Author):

We thank the reviewer for their comments above.